# Pathogenic characteristics of an aggregated diarrhea event caused by *Plesiomonas shigelloides* from stream

**Peng Zhang**[1☯], **Huimin Yao**[2☯], **Lei Ji**[1], **Liping Chen**[1], **Deshun Xu**[1], **Wei Yan**[1]*

1 Huzhou Center for Disease Control and Prevention, Huzhou, China, 2 Anji County Center for Disease Control and Prevention, Huzhou, China

☯ These authors contributed equally to this work.
* yanw1024@163.com

## Abstract

This study aimed to investigate the cause of a foodborne disease outbreak in Huzhou on August 14, 2023. Multiple enteropathogens were detected using FilmArray, and the pathogen was subsequently isolated and cultured from anal swabs of the cases and stream water. The isolated strains were identified using VITEK MS, and antimicrobial susceptibility test, pulsed field gel electrophoresis (PFGE) molecular typing, and whole genome sequencing (WGS) were performed on the isolates of *Plesiomonas shigelloides*. Gene annotation and sequence alignment were used to analyze the virulence genes and drug resistance genes of the strains. A phylogenetic tree was constructed based on single nucleotide polymorphism (SNP), and homology analysis was conducted to trace the origin of *P. shigelloides*. A total of 7 strains of *P.shigelloides* were isolated, with 3 from stream water and 4 from anal swabs. All 7 strains exhibited the same PFGE pattern and showed resistance to amikacin, trimethoprim-sulfamethoxazole, chloramphenicol, tetracycline, cefazolin, streptomycin, and florfenicol. The isolated strains carried the same resistance genes and virulence factors. In the sequences of the isolated strains from this outbreak, 11 mutation sites were detected. The phylogenetic tree based on SNP sites showed that these strains were homologous. This foodborne disease outbreak caused by *P.shigelloides* was the first reported in Huzhou. WGS can be used as a complementary method to PFGE for epidemiological investigations of disease outbreaks.

**Data Availability Statement:** All relevant data are within the manuscript and its Supporting information files.

## Introduction

Foodborne diseases are illnesses caused by consuming contaminated food, with diarrhea and vomiting as the main symptoms. It is a significant global public health issue, with millions of people falling ill each year [1]. *Plesiomonas shigelloides* is a Gram-negative foodborne pathogen that belongs to the family of *Enterobacteriaceae*. Its existence has been known for nearly 80 years, since its discovery by Ferguson in 1947 [2]. It is widely distributed in freshwater rivers, streams, lakes, and has been found in humans, animals, and the environment [3, 4]. The main

**Funding:** This study was funded by Key Laboratory of Emergency Detection for Public Health of Huzhou. This study was supported by grants from "Special support plan for local high-level talents in South Taihu Lake" of Huzhou.

**Competing interests:** The authors have declared that no competing interests exist.

symptom after infection is acute gastroenteritis, and it can also lead to neurological diseases, sepsis, and eye diseases [3]. Compared to other foodborne pathogens such as *Salmonella*, *Vibrio parahemolyticus*, and diarrheal *Escherichia coli*, there have been relatively few reports and less attention given to *P. shigelloides* [5]. On August 14, 2023, a hospital in Anji County of Huzhou reported 7 cases of diarrhea symptoms from the same company. To determine the cause of this outbreak, the CDC of Anji County immediately conducted an investigation and collected relevant specimens for etiological identification. Multiple enteropathogens were detected using FilmArray, and the isolates of *P. shigelloides* underwent antimicrobial susceptibility testing, PFGE molecular typing, and WGS. Based on the results of the epidemiological investigation and laboratory tests, it was concluded that this was an outbreak of diarrhea caused by *P. shigelloides* contamination.

## Materials and methods

### Ethics approval

This study was approved by the human research ethics committee of the Huzhou Center for Disease Control and Prevention. Informed consent for the anal swabs samples was obtained from the patients or their guardians. Verbal consent in the patient's visit notes.

### The outbreak

On August 14, 2023, several patients with acute gastroenteritis were treated at the emergency departments of Anji County People's Hospital, located in Huzhou City, Zhejiang Province, China. These patients reported attending an outdoor team building picnic organized by the company. During the picnic, they consumed fruit that had been washed with stream water and served iced. The suspected outbreak of foodborne diseases was reported to the market supervision department and the health administrative authority. An investigation was subsequently conducted by Anji CDC.

### Case definition

A suspected case was defined as a person with abdominal pain, diarrhea, or vomiting after eating fruits at the picnic in the company since August12.A probable case was determined if diarrhea ($\geq$3 times/24h) or vomiting accompanied by nausea, abdominal pain, fever and other symptoms occurs since August 12.A laboratory-confirmed case could be determined if the anal swab sample from a probable case was tested positive for *P. shigelloides*.

### Epidemiologic investigation

According to the case definition, the local public health physician interviewed the picnic attendees and physician-in-charge, and checked the outpatient records to determine the cases. According to the standardized questionnaire, the possible cases were further investigated. The questionnaire includes basic demographic information (age and sex), clinical information (first symptoms, onset time, frequency of vomiting, abdominal pain, diarrhea, stool characteristics and body temperature) as well as food exposure. On August15, a detailed sanitary investigation was carried out onsite, and stream water were obtained for laboratory testing. There are no kitchen in the company.

### Sample collection and detection

A total of 14 samples were collected in this incident from August 12th to August 14th in 2023, including 11patients and 3 stream water(upstream 1,midstream 1,downstream1). We tested all

collected specimens forFilmArray2.0 PCR analysis system(bioMérieux, France) using Filmarray GI test strip which can detect 22 pathogens, including *P. shigelloides*. After Filmarray, We could determine the scope of bacterial culture. Samples used to test for *P. shigelloides* should be collected and handled the same as those for any routine enteric pathogens. The specimen was inoculated into alkaline peptone water and cultured at 36.0°C for 18 hours. After incubation, the cultures were plated on MacKonkey(MAC), thiosulfate citrate bile salts sucrose agar (TCBS) and Columbia blood agar(Antu, China) for presumptive colony screening. All presumptive colonies were further isolated and identified using the VITEK MS system (bioMérieux, France), according to the manufacturer's instructions.

Antimicrobial susceptibility test Antibiotic susceptibility of all *P. shigelloides* isolates was assayed using the broth micro-dilution minimum inhibitory concentrations (MICs) method according to the Clinical and Laboratory Standards Institute guidelines [6]. Adjust the bacterial suspension to 0.5 McFarland, and add 10 μLbacterial suspension to broth culture medium, then add 50 μL to the antimicrobial susceptibility plate after mixing and seal the membrane, incubate overnight at 36°C.The 29 antibiotics used were ampicillin (AMP), ampicillin/sulbactam (AMS), cefotaxime (CTX), ceftazidime (CAZ), cefoxitin (CFX), cefazolin (CFZ), imipenem (IMP), cefotaxime/clavulanic acid(CTX/C), ceftazidime/clavulanic acid(CAZ/C), amoxicilin/clavulanic acid (AMC), cefuroxime (CXM), cefepime (CPM), ceftazidime/avibactam (CZA), meropenem (MEM), ertapenem(ETP), ceftiofur (CEF), ciprofloxacin (CIP), nalidixic acid (NAL), gentamicin (Gen), amikacin(AMK), streptomycin (STR), tetracycline (TET), tigecycline (TIG), florfenicol (FFC), colistin(CT), polymixin (PB), azithromycin (AZM), trimethoprim-sulfamethoxazole(SXT), and chloramphenicol (CHL). *E. coli* ATCC29522 was used as quality control strain.

## Pulsed field gel electrophoresis

Pulsed field gel electrophoresis (PFGE) was conducted following the method described by Shigematsu [7]. The restriction enzyme used was *Spe*I (TaKaRa, Japan), and the *Salmonellaenterica* serovar Braenderup strain H9812, digested with *Xba*I (Takara, Japan), served as the standard. Electrophoresis was carried out on a CHEF Mapper XA system (Bio-Rad, USA) for a total of 18 hours (with an initial conversion time of 1 s and a final conversion time of 20 s) at an initial current of 135 mA. The gel image was observed after Gelred staining and subsequent decolorization with pure water. Gel images analysis was performed using BioNumerics software v. 7.6 (Applied Maths, Belgium). Clustering was conducted using the unweighted pair group method (UPGMA) and the Dice correlation coefficient, with a position tolerance of 1.5%.

## Whole genome sequencing

The DNA from each strain was extracted from overnight cultures using the QIAamp DNA Mini Kit (Qiagen, Germany), following the manufacturer's instructions. The concentration of the DNA was determined using the Qubit 4 (Thermo, USA), and the qualified DNA was stored at -80°C until further use. WGS libraries were constructed using the Metagenomic DNA Library Kit (Matridx Biotechnology, China) and subsequently sequenced on the NextSeq 550 High Output Reagent Cartridge v2 300 cycles (Illumina, USA).

## Sequence analysis

The quality of the raw data was evaluated using Fastqc software. After quality control, the genome was assembled with Spades. The virulence factor of each strain was determined by referencing the genes deposited in the VFDB database (http://www.mgc.ac.cn/cgi-bin/VFs/v5/

main.cgi). The genomes of *P. shigelloides* were submitted to CARD (https://card.mcmaster.ca/) and the antibiotic resistance genes of the sequenced strains were predicted and analyzed using ABRicate software. SKA was utilized to sort the sequencing results and obtain the core-genome single-nucleotide polymorphisms (SNPs) of the strains. FastTree software was used for sequence alignment and homology analysis of the 44 *P. shigelloides* isolates (with 37 strains from the NCBI GenBank serving as the reference). The resulting phylogenetic tree was visualized using FigTree. BioEdit was employed to calculate the sequence differences among 7 isolates from this outbreak. The genome sequences had been submitted to GenBank. The accession numbers are JAZHQW000000000, JAZHQX000000000, JAZHQY000000000, JAZHQZ000000000,JAZHRA000000000, JAZHRB000000000, JAZHRC000000000.

## Results

### Epidemiological investigation

This study examines an incident involving a company with a total of 18 employees, where 11 cases of illness were reported. The attack rate in this incident was calculated to be 61.11% (11/18). The clinical symptoms observed in the affected individuals included vomiting (100.00%), diarrhea (72.73%), fatigue (36.36%), and fever (36.36%). On August 12, all 18 employees of the company went outside for a picnic. During the picnic, they cooled, rinsed, and consumed fruit soaked in stream water. The fruit was then brought back to the company and eaten on the afternoon of August 13. Interestingly, all the patients had no history of eating together except for the fact that they had consumed the water-soaked fruit. The first case was reported at approximately 6 p.m. on the 13th, presenting with 12 cases of diarrhea and 10 cases of vomiting, but no fever. Out of the affected individuals, 7 sought medical treatment at the hospital, showed improvement after receiving symptomatic treatment, and were subsequently discharged. The remaining 4 cases had mild symptoms and did not require hospitalization. The onset of symptoms occurred between August 13 and August 14, with the shortest incubation period being approximately 7.5 hours, the longest being around 38.8 hours, and an average incubation period of approximately 29.8 hours (Fig 1). Among the 11 cases, 9 were males and 2 were females, with ages ranging from 20 to 26 years.

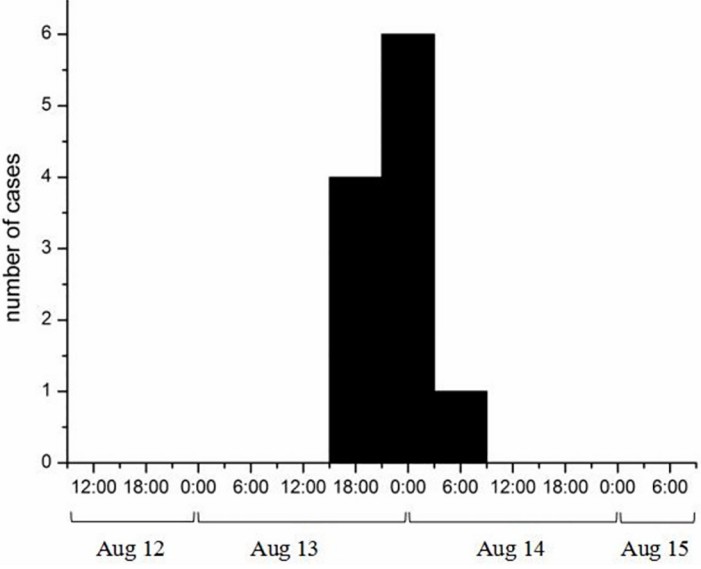

**Fig 1. The onset time distribution of 11 probable cases.**

## Laboratory results

A total of 14 samples were collected, including 11 cases and 3 stream water samples. The pathogen responsible for this event was identified as *P. shigelloides* after initial screening using the Filmarray multi-pathogen detection system. Four strains were derived from patients and three strains were derived from streams. On the MAC plate, the colonies appeared as small pale orange, while on the TCBS plate, they appeared as small green. On the blood plate, the colonies were small and grayish white. A single colony was selected and identified as *P. shigelloides*-positive using VITEK MS.

The seven isolates showed 100% resistance to AMP, SXT, CHL, TET, CFZ, STR, and FFC. It is worth noting that three strains isolated from the stream were resistant to GEN, while the four strains isolated from patients were susceptibility (Table 1).

Further typing of the seven *P. shigelloides* isolates was conducted using PFGE. The PFGE cluster analysis revealed that the isolates from anal swabs of four patients and three stream samples had exactly the same PFGE band type, indicating 100% homology (Fig 2).

The comparison of the CARD database revealed that 7 strains carried 5 identical drug resistance genes. These genes include *adeF* (fluoroquinolone antibiotic, tetracycline antibiotic), *KpnH* (macrolide antibiotic, fluoroquinolone antibiotic, aminoglycoside antibiotic, carbapenem, cephalosporin, penam, peptide antibiotic, penem), *rsmA* (macrolide antibiotic, fluoroquinolone antibiotic, aminoglycoside antibiotic, carbapenem, cephalosporin, penam, peptide antibiotic, penem), *CRP* (macrolide antibiotic, fluoroquinolone antibiotic, penam), and *Haemophilus influenzae PBP3* (conferring resistance to beta-lactam antibiotics such as cephalosporin, cephamycin, penam) and theyareantibiotic efflux pumps. The findings were consistent with the results of the antimicrobial susceptibility test. All 7 strains carried the same virulence genes, except for S2023592 isolated from upstream water, which did not contain *cgsE* (Fig 3). The SNPs clustering analysis, based on the genome sequences of 44 *P. shigelloides* isolates (37 strains from the NCBI GenBank as the reference and 7 strains from this event), showed that the sequence of the 7 strains in this case was identical (Fig 4). When compared to the reference sequences, the 7 strains were most similar to the reference gene sequence "GCF_959021465.1_DRR221358_bin.47_MetaWRAP_v1.3_MAG_genomic", which was submitted by EMG in June 2023. The analysis of sequence differences among the 7 isolates revealed a range of 2 to 11 variations (Table 2).

## Discussion

A total of 18 specimens were collected in this incident, and 7 strains of *P. shigelloides* were detected from all 14 specimens in this aggregated diarrhea event. The event was found to be caused by consuming fruit contaminated with *P. shigelloides*, as confirmed by epidemiological investigation and laboratory results. This food-borne outbreak caused by *P. shigelloides* is the first in our city and province, although similar outbreaks have been reported both domestically and internationally [7, 8]. In our country, food-borne outbreaks are mainly attributed to pathogenic microorganisms such as *Salmonella* and *Vibrio parahaemolyticus* [9, 10]. However, there have been limited reports, both domestically and internationally, on food-borne outbreaks caused by *P. shigelloides* [3]. The prevalence of *P. shigelloides* enteritis varies significantly, with higher rates reported in Africa and Southeast Asia compared to lower numbers in Europe and North America. Most gastrointestinal infections caused by *P. shigelloides* result from consuming undercooked food, though the majority of infections are mild and do not require antimicrobial intervention or other medical treatments. A large study on Plesiomonas diarrhea found that 85% of cases were self-limiting [11]. However, for moderate to severe cases of diarrhea, antimicrobial therapy may be necessary. DuPont recommends ciprofloxacin

**Table 1. Antibiotic resistance of *P. shigelloides*.**

| Key | β-lactam | | | | | | | | | | | | | | | | Quinolones | | Aminoglycosides | | | Tetracycline | | | lipopeptides | | Macrolides | Folate antagonists | Phenylpropanols |
|---|---|---|---|---|---|---|---|---|---|---|---|---|---|---|---|---|---|---|---|---|---|---|---|---|---|---|---|---|---|
| | AMP | AMS | CTX | CAZ | CFX | CFZ | IPM | CTX/C | CAZ/C | AMC | CXM | CPM | CZA | MEM | ETP | CEF | CIP | NAL | GEN | AMK | STR | TET | TIG | FFC | CT | PB | AZM | SXT | CHL |
| S2023594 (down) | R | S | S | S | S | R | S | S | S | S | S | S | S | S | S | S | S | S | R | S | R | R | S | R | I | I | S | R | R |
| S2023593 (mid) | R | S | S | S | S | R | S | S | S | S | S | S | S | S | S | S | S | S | R | S | R | R | S | R | I | I | S | R | R |
| S2023592 (up) | R | S | S | S | S | R | S | S | S | S | S | S | S | S | S | S | S | S | R | S | R | R | S | R | I | I | S | R | R |
| S2023591 | R | S | S | S | S | R | S | S | S | S | S | S | S | S | S | S | S | S | S | S | R | R | S | R | I | I | S | R | R |
| S2023590 | R | S | S | S | S | R | S | S | S | S | S | S | S | S | S | S | S | S | S | S | R | R | S | R | I | I | S | R | R |
| S2023589 | R | S | S | S | S | R | S | S | S | S | S | S | S | S | S | S | S | S | S | S | R | R | S | R | I | I | S | R | R |
| S2023588 | R | S | S | S | S | R | S | S | S | S | S | S | S | S | S | S | S | S | S | S | R | R | S | R | I | I | S | R | R |

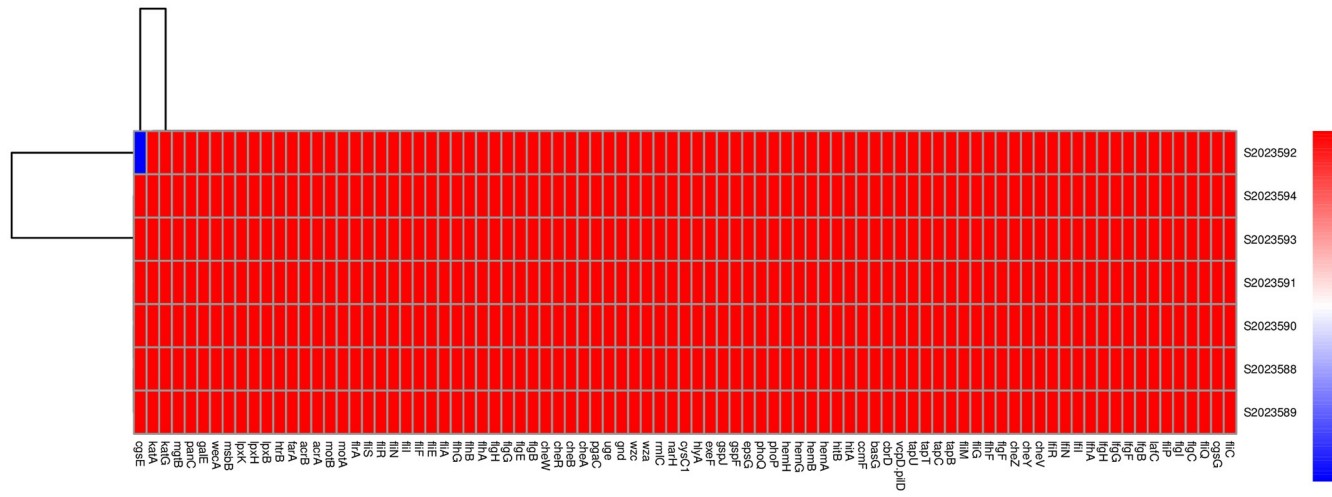

PFGE    PFGE-SpeI

| key | durg resistance | origin |
|---|---|---|
| S2023588 | AMP-SXT-CHL-TET-CFZ-STR-FFC | patient |
| S2023589 | AMP-SXT-CHL-TET-CFZ-STR-FFC | patient |
| S2023590 | AMP-SXT-CHL-TET-CFZ-STR-FFC | patient |
| S2023591 | AMP-SXT-CHL-TET-CFZ-STR-FFC | patient |
| S2023592 | AMP-SXT-CHL-GEN-TET-CFZ-STR-FFC | upstream water |
| S2023593 | AMP-SXT-CHL-GEN-TET-CFZ-STR-FFC | midstream water |
| S2023594 | AMP-SXT-CHL-GEN-TET-CFZ-STR-FFC | downstream water |

**Fig 2. PFGE pattern of *P. shigelloides* isolates from stream and patients.**

or azithromycin as treatment options for severe cases of dysentery-like diarrhea caused by *P. shigelloides* [12]. It is worth noting that *P. shigelloides* is not included in our National food-borne disease surveillance, and its clinical symptoms are similar to those caused by other food-borne pathogens. As a result, infections caused by *P. shigelloides* are often underestimated and receive insufficient attention [13].

*P. shigelloides* is a Gram-negative enterobacterium and a zoonotic pathogen [14]. It is commonly associated with intestinal infections caused by water contamination [15]. *P. shigelloides* can be found in ponds, rivers, streams, and aquatic animals, particularly fish, which serve as its primary host [16, 17]. Greenlees *et al.* [18] conducted a study that revealed a higher incidence of reported cases during the warmer months when water temperatures increase, facilitating the proliferation of plesiomonads through sewage contamination. This event took place in August, one of the hottest months of the year, which aligns with Greenlees's findings. The frequency of plesiomonads as enteric pathogens recovered from diarrheal stools in tropical or subtropical regions ranges from 2% to 10% [19–21]. The frequency in patients with diarrhea of Huzhou was 2.32%(24/1034, exclude outbreak) in 2023. The test of *P. shigelloides* will be added to Huzhou foodborne disease surveillance in 2024. Furthermore, another study reported a significant decrease in the frequency of stool pathogens after practicing personal hygiene [22]. In addition to contact with or consumption of food contaminated with *P. shigelloides*, risk factors also include foreign travel [7], individuals with weakened immune systems [23], low socioeconomic status, and poor hygiene conditions [24], among others.

**Fig 3. Virulence factors detected in the examined 7 *P. shigelloides* isolates in this study.**

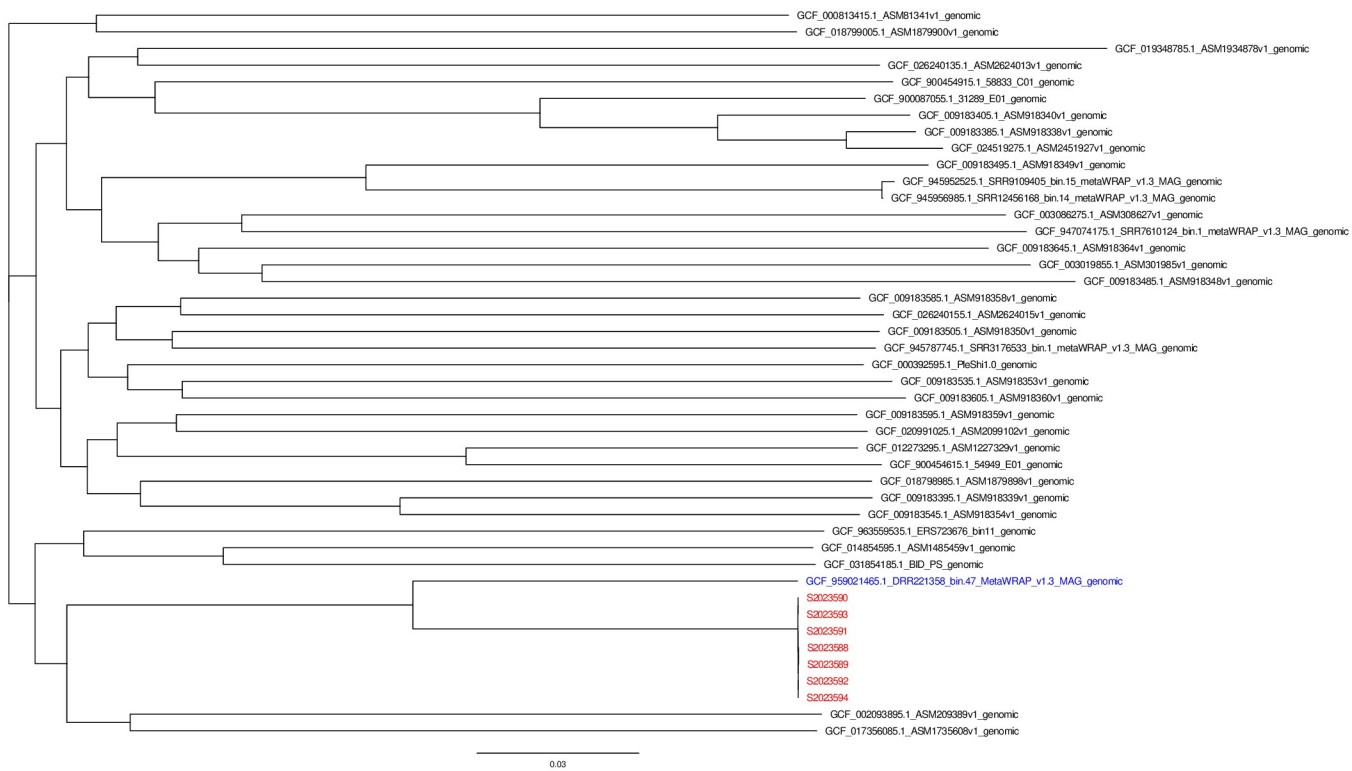

**Fig 4. Phylogenetic tree based on SNPs of 44 *P. shigelloides*.**

The use of antibiotics is necessary for patients with severe diarrhea. DuPont's recommendation of ciprofloxacin or azithromycin [12] aligns with our study, which found that 7 strains were sensitive to both antibiotics. Our results on drug resistance revealed that these 7 strains were susceptibility to most antibiotics but resistant to AMP, CFZ, TET, and CHL, consistent with findings from related studies [25, 26]. The multiple drug resistance(MDR)percentage of *P. shigelloides* strains in our study was 100.00%, which will bring difficulties to further clinical treatment, because the delay in administration of active antimicrobial therapy may lead to poor outcomes. Notably, *P. shigelloide* strains from the stream and patients displayed different resistance and susceptibility patterns to GEN (stream resistance, patient susceptibility), possibly due to variations in internal and external environments.

PFGE is considered the gold standard for outbreak tracing due to its high resolution, good repeatability, stable results, and easy standardization [27]. It is helpful in timely detecting

**Table 2. The sequence difference of 7 isolates.**

| Seq-> | S2023588 | S2023589 | S2023590 | S2023591 | S2023592 | S2023593 | S2023594 |
|---|---|---|---|---|---|---|---|
| S2023588 | 0 | 5 | 8 | 7 | 6 | 10 | 5 |
| S2023589 | 5 | 0 | 7 | 6 | 9 | 11 | 4 |
| S2023590 | 8 | 7 | 0 | 3 | 8 | 4 | 3 |
| S2023591 | 7 | 6 | 3 | 0 | 5 | 7 | 2 |
| S2023592 | 6 | 9 | 8 | 5 | 0 | 8 | 5 |
| S2023593 | 10 | 11 | 4 | 7 | 8 | 0 | 7 |
| S2023594 | 5 | 4 | 3 | 2 | 5 | 7 | 0 |

outbreaks caused by *P. shigelloide* and controlling foodborne diseases. Based on the results of PFGE, the *P. shigelloide* strains were clustered together, indicating that the strains from both the stream and the patient are homologous. Therefore, it can be inferred that this incident was caused by consuming contaminated fruit from the stream.

With the decreasing cost of sequencing, whole-genome sequencing (WGS) has gradually been employed for monitoring foodborne pathogens. Among the 7 strains analyzed, they all carried the same drug resistance genes, including *adeF*, *KpnH*, *rsmA*, *CRP*, and *PBP3*. These resistance genes confer resistance to various antibiotics such as quinolones, fluoroquinolones, tetracycline, macrolides, aminoglycosides, lipopeptides, phenylpropanols, and β-lactam antibiotics. The results of the antimicrobial susceptibility test were consistent with these findings. Efflux of drug via an efflux pump is a key resistance mechanism in Enterobacteriaceae strains. *rsmA*, *rsmA*, *adeF* and *CRP* are the classic resistance-nodulation-cell division (RND) efflux pump, and *KpnH* is major facilitator superfamily (MFS)efflux pump. The overexpression of efflux pumps is a potential resistance mechanism in *P. shigelloide*. The virulence genes were also completely consistent among the strains, except for S2023592, which did not contain cgsE. The absence of *cgsE* indicates a slight variation in adherence factors within the strains, while overall, the 7 strains showed high homology. *cgsE* belongs to the curli fibers of the adherence factor family.

SNP analysis based on whole-genome sequencing (WGS) can be utilized as a novel molecular typing method for outbreak tracing [28]. By constructing a SNP phylogenetic tree, we observed that the pathogens involved in this event belonged to the same evolutionary branch and exhibited high homology. The most closely related strain was GCF_959021465.1_DRR221358_bin.47_MetaWRAP_v1.3_MAG_genomic, which was submitted in June 2023, just two months before the occurrence of this aggregated diarrhea event. The sequence was collected from human feces in the United Kingdom. While this reference sequence is the most closely related to our study, there are still 18225 differences in SNP sites. Therefore, we cannot assume that they are homologous. Bioedit analysis revealed a maximum of 11 SNP variation sites among the 7 strains. It is generally accepted that less than 20 SNP variation sites indicate homology.

The detection rate of *P. shigelloide* was found to be highest during summer and lowest during winter. The detection rate in sentinel hospitals was generally low [26], making *P. shigelloide* easy to overlook. However, in this case, the use of Filmarray allowed for determination of the range before isolation and culture, enabling isolation of the positive strain. Although PFGE remains the primary method for outbreak tracing, WGS used in this investigation, can more accurately identify the outbreak strain and identify the closest relatives worldwide. Furthermore, WGS can provide information on drug resistance genes, virulence genes, and SNP mutation sites of strains, thereby enhancing pathogen monitoring. This serves as an effective complement to PFGE. As new laboratory techniques continue to develop, outbreak tracing analysis will become faster and more accurate.

## Conclusions

This is an aggregation diarrhea event caused by *P. shigelloide* after PFGE and WGS analysis. All isolates carry the same drug resistance genes and virulence genes except *cgsE*, and the homology between isolates is high.

## Supporting information

**S1 Raw images.**
(PDF)

**S1 Data.**
(NEWICK)

## Acknowledgments

We thank the staff of the Anji County CDC for collecting the samples and providing epidemiological data.

## Author Contributions

**Conceptualization:** Peng Zhang, Wei Yan.

**Data curation:** Deshun Xu.

**Formal analysis:** Huimin Yao, Lei Ji.

**Investigation:** Huimin Yao.

**Methodology:** Liping Chen.

**Project administration:** Peng Zhang.

**Software:** Wei Yan.

**Validation:** Lei Ji, Wei Yan.

**Writing – original draft:** Huimin Yao, Wei Yan.

**Writing – review & editing:** Peng Zhang, Lei Ji.

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
