## [Decision Letter · Decision Letter 0]

5 Mar 2024

PONE-D-24-02504Pathogenic characteristics of an aggregated diarrhea event caused by Pseudomonas shigelliides from streamPLOS ONE

Dear Dr. Yan,

Thank you for submitting your manuscript to PLOS ONE. After careful consideration, we feel that it has merit but does not fully meet PLOS ONE’s publication criteria as it currently stands. Therefore, we invite you to submit a revised version of the manuscript that addresses the points raised during the review process.

Please review the feedback from the Reviewers carefully and submit the revised version/responses accordingly.

We look forward to receiving your revised manuscript.

Kind regards,

Furqan Kabir

Academic Editor

PLOS ONE

Reviewers' comments:

Reviewer's Responses to Questions

**Comments to the Author**

1. Is the manuscript technically sound, and do the data support the conclusions?

Reviewer #1: Yes

Reviewer #2: Yes

2. Has the statistical analysis been performed appropriately and rigorously? 

Reviewer #1: Yes

Reviewer #2: Yes

3. Have the authors made all data underlying the findings in their manuscript fully available?

Reviewer #1: Yes

Reviewer #2: Yes

4. Is the manuscript presented in an intelligible fashion and written in standard English?

Reviewer #1: Yes

Reviewer #2: Yes

5. Review Comments to the Author

Reviewer #1: Reviewer's Comments on the Manuscript:

The paper studied combines epidemiological investigation with laboratory results, analyzing an aggregation diarrhea event which caused by Pseudomonas shigelliides. Pseudomonas shigelliides as an enteropathogen are often underestimated and receive insufficient attention in the world. Most reports of P. shigelliides outbreaks are more years old. This study isolated P. shigelliides from patients and water samples, then analyzing pathogenic characteristics of the strains by drug sensitivity test, PFGE and WGS. The study has a good indicative effect for food contaminated by P. shigelliides in water. But there are some obstacles for the readers to understand, and the following comments should be considered in a revision.

1. Abstract, the content description is inaccurate. The author should state the time of event clearly, because there may be not only one aggregation diarrhea event in Huzhou,2023. It is suggested to describe it as “trace the origin of P. shigelliides” rather than “trace the origin of the pathogenic bacteria”. It is suggested to delete the "of the cases" in line 12.

2.The English expression in the paper needs to be consistent for clarity. After the first appearance of "Pseudomonas shigella", "P. shigelliides" can be used as a substitute in the following. The same to "WGS" replace " whole genome sequencing".

3. Sample collection and detection, it should be noted that which 22 pathogens can be detected by Filmarray GI test strip.

4. Sample collection and detection, which paper or methods were referenced for the isolation and culture. P. shigelliides does not have standards for isolation now.

5. Sequence analysis, the genome sequences of the 7 P. shigelloides isolates should be submitted to GenBank/DDBJ/ENA and acquired the accession numbers.

6. Discussion, in paragraph 1, it is recommended to add a description of local distribution of P. shigelliides formerly and the measures taken by the CDC after this diarrhea aggregation event.

7. Discussion, antibiotic resistance, especially multidrug resistance(MDR), is becoming increasingly severe. It is suggested to add discussion of MDR in paragraph 3.

8. Discussion, in paragraph 6,this study found the most closely related strain was GCF_959021465.1_DRR221358_bin.47_MetaWRAP_v1.3_MAG_genomic. But there is no description of the differences in the number of SNP variation. It is suggested to supplement the explanation in the paper.

Overall, the research holds promise and contributes meaningfully to the field. It is one of the few reports in recent years regarding the outbreak of P. shigelloides. In addition to patients, strains were also isolated from the stream water that is important for studying of P. shigelloides. However, addressing the aforementioned concerns and making the suggested modifications will undoubtedly strengthen the paper, ensuring its readiness. It is suggested to publish after modification.

Reviewer #2: 1. Please verify, it is ‘Pseudomonas shigelliides’ or ‘Pseudomonas shigelloides’ or ‘Plesiomonas shigelloides’. In all the case it seems not ‘Pseudomonas’ See this article : Janda JM, Abbott SL, McIver CJ. Plesiomonas shigelloides Revisited. Clin Microbiol Rev. 2016;29(2):349-74. REALLY IT IS NOT ALLOWED TO MAKE THIS MISTAKES OF CONFUSION.

2. Please correct ‘Gram-negative’ not ‘gram-negative'. Verify this in all your article (see section discussion also)

3. In ‘diarrheal Escherichia coli’, the term ‘diarrheal’ should not be in italic.

4. In section ‘Sample collection and detection’, is this correct ‘pathogens, including Pseudomonas shigella’? please verify all your manuscript according to my first comment.

5. The term ‘MAC’ means ‘MacKonkey’ ? please write the full term without abbreviation, at least for the first time that you mention it.

6. Change the term’ Drug sensitivity test’ by ‘antimicrobial susceptibility test’

7. In all the manuscript, please verify punctuation and there are a lot of word not separated (especially in the section ‘antimicrobial susceptibility test’. Please never write the name of antibiotics with capital letters, write ‘ampicillin (AMP), ampicillin/sulbactam (AMS), cefotaxime (CTX), ceftazidime (CAZ), cefoxitin (CFX), Cefazolin (CFZ), imipenem (IMP), cefotaxime/clavulanic acid (CTX/C), ceftazidime/clavulanic acid (CAZ/C), amoxicilin/clavulanic acid (AMC), cefuroxime (CXM), cefepime (CPM), ceftazidime/avibactam (CZA), meropenem (MEM), ertapenem(ETP), ceftiofur (CEF), ciprofloxacin (CIP), nalidixic acid (NAL), gentamicin (Gen), amikacin(AMK), streptomycin (STR), tetracycline (TET), tigecycline (TIG), florfenicol (FFC), colistin (CT), Polymixin (PB), azithromycin (AZM), trimethoprim-sulfamethoxazole (SXT), and chloramphenicol (CHL). E. coli ATCC29522 was used as quality control strain’. THE ‘E. coli’ must be in italic.

8. The name of the enzyme used in PFGE must be in italic ‘Spe’ and ‘I’ not in italic, this is also for ‘Xba’

9. In ‘Salmonellaenterica serovar Braenderup’ make space between ‘Salmonella’ and enterica’ and write them in italic.

10. Please in section results, mention that kpnH, rsmA, rsmA, adeF and CRP are ‘antibiotic efflux pumps’ or ‘efflux resistance genes’. This could explain the multidrug resistance phenotype caused by each pump.

11. Please correct as follows:’.. Greenlees et al. [18]conducted a study that revealed a higher incidence of reported cases during the warmer months when water temperatures increase, facilitating the proliferation of plesiomonads through sewage contamination.’

12. Please in discussion, add 3 to 4 sentences describing the antibiotic efflux pumps (kpnH, rsmA, rsmA, adeF, CRP).

13. You said ‘which was submitted in June 2023,’ Please say from which country this strain was collected, and its source (food, water, patient???).

14. In conclusion, add ‘s” (several genes), so write ‘All isolates carry the same drug resistance genes and virulence genes except cgsE , and the homology between isolates is high.’

6. PLOS authors have the option to publish the peer review history of their article (what does this mean?). If published, this will include your full peer review and any attached files.

Reviewer #1: **Yes: **Qingli Dong

Reviewer #2: **Yes: **Mohamed Salah Abbassi

---

## [Author Response · Author response to Decision Letter 0]

11 Mar 2024

Dear Reviewer and Editor:

We are very grateful to Reviewer for reviewing the paper so carefully, and hope that the correction will meet with approval. We have tried our best to improve and made some changes in the manuscript. Thank you very much for your comments and suggestions.

Reviewer #1:

1. Abstract, the content description is inaccurate. The author should state the time of event clearly, because there may be not only one aggregation diarrhea event in Huzhou,2023. It is suggested to describe it as “trace the origin of P. shigelliides” rather than “trace the origin of the pathogenic bacteria”. It is suggested to delete the "of the cases" in line 12.

Response: Thanks for your advice. The statements have been corrected. We will be happy to edit the text further based on comments from reviewers.

2.The English expression in the paper needs to be consistent for clarity. After the first appearance of "Pseudomonas shigella", "P. shigelliides" can be used as a substitute in the following. The same to "WGS" replace " whole genome sequencing".

Response: Thanks for your advice. These statements have been corrected in all the manuscript.

3. Sample collection and detection, it should be noted that which 22 pathogens can be detected by Filmarray GI test strip. 

Response: Thanks for your advice. This Filmarray GI test strip include Campylobacter (C. jejuni/C. coli/ C. upsaliensis), Clostridium difficile (toxin A/B), Plesiomonas shigelloides, Salmonella, Vibrio (V. parahaemolyticus/V. vulnificus/ V. cholerae), Yersinia enterocolitica, EAEC, EPEC, ETEC, STEC, E. coli O157, EIEC, Cryptosporidium, Cyclospora cayetanensis, Entamoeba histolytica, Giardia lamblia (also known as G. intestinalis and G. duodenalis), Adenovirus F 40/41, Astrovirus, Norovirus GI/GII, Rotavirus A, Sapovirus (Genogroups I, II, IV, and V). 4. Sample collection and detection, which paper or methods were referenced for the isolation and culture. P. shigelliides does not have standards for isolation now.

Response: Thanks for your advice. Samples used to test for Plesiomonas shigelloides should be collected and handled the same as those for any routine enteric pathogens. According to reference 3 to isolation and culture. 

5. Sequence analysis, the genome sequences of the 7 P. shigelloides isolates should be submitted to GenBank/DDBJ/ENA and acquired the accession numbers.

Response: Thanks for your advice. The genome sequences of the 7 P. shigelloides isolates had been submitted to GenBank. The accession numbers are JAZHQW000000000, JAZHQX000000000, JAZHQY000000000, JAZHQZ000000000, JAZHRA000000000, JAZHRB000000000, JAZHRC000000000.

6. Discussion, in paragraph 1, it is recommended to add a description of local distribution of P. shigelliides formerly and the measures taken by the CDC after this diarrhea aggregation event.

Response: Thanks for your advice. These statements have been added in discussion.

7. Discussion, antibiotic resistance, especially multidrug resistance(MDR), is becoming increasingly severe. It is suggested to add discussion of MDR in paragraph 3.

Response: Thanks for your advice. We agree with you and added some sentences about MDR in discussion paragraph 3.

8. Discussion, in paragraph 6,this study found the most closely related strain was GCF_959021465.1_DRR221358_bin.47_MetaWRAP_v1.3_MAG_genomic. But there is no description of the differences in the number of SNP variation. It is suggested to supplement the explanation in the paper.

Response: Thanks for your advice. Although this reference sequence is the most closely related with our study, there are still 18225 differences in SNP sites. We cannot assume that they are homologous. These statements have been added in discussion.

Reviewer #2:

1. Please verify, it is ‘Pseudomonas shigelliides’ or ‘Pseudomonas shigelloides’ or ‘Plesiomonas shigelloides’. In all the case it seems not ‘Pseudomonas’ See this article : Janda JM, Abbott SL, McIver CJ. Plesiomonas shigelloides Revisited. Clin Microbiol Rev. 2016;29(2):349-74. REALLY IT IS NOT ALLOWED TO MAKE THIS MISTAKES OF CONFUSION.

Response: We thank the reviewer of pointing out this issue. We apologize for the confusion generated by the manuscript. It is Plesiomonas shigelloides in this article. We had replaced all ‘Pseudomonas shigelliides’ with ‘Plesiomonas shigelloides’.

2. Please correct ‘Gram-negative’ not ‘gram-negative'. Verify this in all your article (see section discussion also)

Response: We are very sorry for our incorrect writing and it is rectified.

3. In ‘diarrheal Escherichia coli’, the term ‘diarrheal’ should not be in italic.

Response: Thanks for your advice. We are very sorry for our incorrect writing and it is rectified.

4. In section ‘Sample collection and detection’, is this correct ‘pathogens, including Pseudomonas shigella’? please verify all your manuscript according to my first comment.

Response: Sorry, we had replaced ‘Pseudomonas shigelliides’ with ‘Plesiomonas shigelloides’ in my manuscript.

5. The term ‘MAC’ means ‘MacKonkey’ ? please write the full term without abbreviation, at least for the first time that you mention it.

Response: Thanks for your advice. The term ‘MAC’ means ‘MacKonkey’. We had changed to the full term.

6. Change the term’ Drug sensitivity test’ by ‘antimicrobial susceptibility test’.

Response: Thanks for your advice. We had replaced ‘Drug sensitivity test’ with ‘antimicrobial susceptibility test’ .

7. In all the manuscript, please verify punctuation and there are a lot of word not separated (especially in the section ‘antimicrobial susceptibility test’. Please never write the name of antibiotics with capital letters, write ‘ampicillin (AMP), ampicillin/sulbactam (AMS), cefotaxime (CTX), ceftazidime (CAZ), cefoxitin (CFX), Cefazolin (CFZ), imipenem (IMP), cefotaxime/clavulanic acid (CTX/C), ceftazidime/clavulanic acid(CAZ/C), amoxicilin/clavulanic acid (AMC), cefuroxime (CXM), cefepime (CPM), ceftazidime/avibactam (CZA), meropenem (MEM), ertapenem(ETP), ceftiofur (CEF), ciprofloxacin (CIP), nalidixic acid (NAL), gentamicin (Gen), amikacin(AMK), streptomycin (STR), tetracycline (TET), tigecycline (TIG), florfenicol (FFC), colistin (CT), polymixin (PB), azithromycin (AZM), trimethoprim-sulfamethoxazole (SXT), and chloramphenicol (CHL).E. coli ATCC29522 was used as quality control strain’. THE ‘E. coli’ must be in italic.

Response: Thanks for your advice. We had verified punctuation in all the manuscript and edited the capital letters of the antibiotics. The ‘E. coli’ was used in italic.

8. The name of the enzyme used in PFGE must be in italic ‘Spe’ and ‘I’ not in italic, this is also for ‘Xba’

Response: We are very sorry for our incorrect writing and it is rectified.

9. In ‘Salmonellaenterica serovar Braenderup’ make space between ‘Salmonella’ and enterica’ and write them in italic.

Response: Thanks for your advice and the statement have been corrected.

10. Please in section results, mention that kpnH, rsmA, rsmA, adeF and CRP are ‘antibiotic efflux pumps’ or ‘efflux resistance genes’. This could explain the multidrug resistance phenotype caused by each pump.

Response: Thanks for your advice. kpnH, rsmA, rsmA, adeF and CRP are ‘antibiotic efflux pumps’. These were added in section results.

11. Please correct as follows:’.. Greenlees et al. [18]conducted a study that revealed a higher incidence of reported cases during the warmer months when water temperatures increase, facilitating the proliferation of plesiomonads through sewage contamination.’

Response: Thanks for your advice. We had corrected as follows: ‘.. Greenlees et al. [18]conducted a study that revealed a higher incidence of reported cases during the warmer months when water temperatures increase, facilitating the proliferation of plesiomonads through sewage contamination.’

12. Please in discussion, add 3 to 4 sentences describing the antibiotic efflux pumps (kpnH, rsmA, rsmA, adeF, CRP).

Response: Thanks for your advice. We had added some sentences describing the antibiotic efflux pumps in discussion.

13. You said ‘which was submitted in June 2023,’ Please say from which country this strain was collected, and its source (food, water, patient???).

Response: The RefSeq of this strain was collected from United Kingdom and its source was human feces. These were added to the manuscript. We can found information from ‘https://www.ncbi.nlm.nih.gov/nuccore/CAUEVM000000000.1’

14. In conclusion, add ‘s” (several genes), so write ‘All isolates carry the same drug resistance genes and virulence genes except cgsE , and the homology between isolates is high.’

Response: Thanks for your advice. We are very sorry for our incorrect writing and it is rectified.

---

## [Editor Report · Decision Letter 1]

19 Mar 2024

Pathogenic characteristics of an aggregated diarrhea event caused by Plesiomonas shigelloides from stream

PONE-D-24-02504R1

Dear Dr. Yan,

We’re pleased to inform you that your manuscript has been judged scientifically suitable for publication and will be formally accepted for publication once it meets all outstanding technical requirements.

An invoice for payment will follow shortly after the formal acceptance. To ensure an efficient process, please log into Editorial Manager at Editorial Manager® , click the 'Update My Information' link at the top of the page, and double check that your user information is up-to-date. If you have any billing related questions, please contact our Author Billing department directly at authorbilling@plos.org.

Kind regards,

Furqan Kabir

Academic Editor

PLOS ONE
---

## [Editor Report · Acceptance letter]

26 Mar 2024

PONE-D-24-02504R1 

PLOS ONE

Dear Dr. Yan, 

I'm pleased to inform you that your manuscript has been deemed suitable for publication in PLOS ONE. Congratulations! Your manuscript is now being handed over to our production team.

Kind regards, 

on behalf of

Dr. Furqan Kabir 

Academic Editor

PLOS ONE